# *Salmonella* Typhimurium biofilm disruption by a human antibody that binds a pan-amyloid epitope on curli

Sarah A. Tursi[1], Rama Devudu Puligedda[2], Paul Szabo[3], Lauren K. Nicastro[1], Amanda L. Miller[1], Connie Qiu [1], Stefania Gallucci[1], Norman R. Relkin[3], Bettina A. Buttaro[1], Scott K. Dessain[2,4✉] & Çagla Tükel [1,4✉]

Bacterial biofilms, especially those associated with implanted medical devices, are difficult to eradicate. Curli amyloid fibers are important components of the biofilms formed by the *Enterobacteriaceae* family. Here, we show that a human monoclonal antibody with pan-amyloid-binding activity (mAb 3H3) can disrupt biofilms formed by *Salmonella enterica* serovar Typhimurium in vitro and in vivo. The antibody disrupts the biofilm structure, enhancing biofilm eradication by antibiotics and immune cells. In mice, 3H3 injections allow antibiotic-mediated clearance of catheter-associated *S.* Typhimurium biofilms. Thus, monoclonal antibodies that bind a pan-amyloid epitope have potential to prevent or eradicate bacterial biofilms.

[1] Department of Microbiology and Immunology, Lewis Katz School of Medicine, Temple University, Philadelphia, PA, USA. [2] Lankenau Institute for Medical Research, Wynnewood, PA, USA. [3] Weill Cornell Medical Feil Family Brain & Mind Research Institute, New York, NY, USA. [4]These authors contributed equally: Scott K. Dessain, Çagla Tükel. ✉email: dessains@mlhs.org; ctukel@temple.edu

Biofilms are three-dimensional multicellular communities that allow bacteria to irreversibly adhere to indwelling medical devices and provide resistance to antibiotics and prevent eradication by innate immune cells. Biofilm-associated infections of indwelling medical devices are refractory to antibiotic treatment and require surgical debridement and/or device removal[1–7]. Biofilm-producing enteric bacteria including *Salmonella enterica* and *Escherichia coli* remain to be the major cause of many bloodstream infections[8–11]. While biofilms of *Salmonella* play a critical role in persistent infections[11], *E. coli* biofilms are important causes of prosthetic joint infections, recurrent urinary tract infections, and central line-associated blood infections (CLABSIs)[12–16].

Biofilms contribute to the development antibiotic tolerance and resistance[17–19]. Biofilms form a physical barrier to many antibiotics, restricting effective drug concentrations to sub-lethal levels and promoting the outgrowth of resistant strains[20]. They also block host immune responses, protecting cells from phagocytosis and complement mediated killing[21–23]. Lastly, biofilms enhance the ability of bacteria to generate "persister" sub-populations, which are multi-drug tolerant, essentially dormant cells with a characteristic gene expression signature[24]. Dispersal of a biofilm has the potential to seed a patient with drug-resistant bacteria[25,26]. Strategies are needed to break down biofilms associated with indwelling medical devices and chronic infections, in order to render bacteria sensitive to antibiotics and susceptible to immune clearance.

Curli amyloid fibers are the major constituent (approximately 85%) of the extracellular matrix in biofilms formed by the members of *Enterobacteriaceae* family, e.g. *E. coli* and *Salmonella* ssp.[27–29]. Curli is a heteropolymeric amyloid fibril comprising two subunits, CsgA and CsgB, in a ~20:1 ratio. CsgB nucleates CsgA fibril formation in the biofilm and attaches the amyloid fibrils to the bacterial surface[30–34]. Curli also promotes adhesion among bacteria within the biofilm and aids in surface attachment[35,36]. Amyloids such as curli form insoluble protein polymers that are characterized by a repeated cross-beta-sheet structure that can display conformational antibody-binding epitopes shared by amyloids of completely unrelated protein sequences[37]. Monoclonal antibodies (mAbs) that bind these conformational epitopes can inhibit polymerization of amyloidogenic proteins and disperse aggregated amyloids under some conditions.

3H3 is a human mAb that preferentially binds amyloid beta protein (Aβ) oligomers and fibrils, relative to Aβ monomers, as well as many types of pathologic human amyloids, including immunoglobulin light chain and transthyretin amyloids[38]. 3H3 inhibits polymerization of Aβ and other amyloid precursors in vitro and reduces amyloid deposition in transgenic mouse models of AD and familial Danish dementia.

In this study, we investigate whether the 3H3 mAb has analogous effects against curli amyloids, using *S*. Typhimurium biofilms as an experimental model. We show that 3H3 inhibits the polymerization of curli, leading to alteration of the biofilm architecture and rendering the biofilm bacteria more sensitive to antibiotic treatment and to macrophage uptake. In addition, 3H3 inhibits biofilm formation on a vascular catheter in vivo and collaborates with an antibiotic to clear an established, catheter-associated biofilm.

## Results

### Anti-Aβ mAb inhibits formation of *S*. Typhimurium biofilms.
We tested the 3H3 mAb for activity against *S*. Typhimurium biofilm formation in vitro. We compared its activity to the 6A isotype control IgG[39] and three additional human mAbs, 4A6, 4G1, and 2C10, which were obtained from an Alzheimer's disease patient and preferentially bind to oligomeric Aβ forms (Supplementary Table 1). The 3H3 has been previously observed to bind a pan-amyloid epitope. Of the new mAbs, only 4G1 showed potential pan-amyloid binding, as it and the 3H3 mAb both also bind to tau-paired helical filaments (PHF) (Supplementary Fig. 1). 4G1 also inhibits Aβ fibril formation in vitro (Supplementary Fig. 2). As a negative control for biofilm formation, we used the isogenic *S*. Typhimurium mutant, *csgBA*, which does not express curli[40]. We cultured *S*. Typhimurium under biofilm-inducing conditions on glass coverslips at 28 °C for 72 h in the presence of 3H3, then rigorously washed the cultures and stained them with Syto9 green nucleic acid stain (bacteria) and Congo Red (CR) amyloid stain. We imaged the biofilms using confocal scanning laser microscopy.

Three-dimensional reconstructions of the z-stacks showed that the untreated *S*. Typhimurium biofilm was highly structured and covered most of the surface with an average thickness of 20 μm (Fig. 1a), in comparison to the *csgBA* strain, which had no biofilm. The biofilm exposed to the control 6A antibody was similar to the untreated control, 18.21 μm thick and essentially confluent, although the depth of CR staining was reduced. Biofilms formed in cultures containing the other mAbs were substantially thinner, approaching the value for the *csgBA* mutant (5.5 μm): 4A6, 10.9 μm; 4G1, 7.3 μm; 2C10, 9.5 μm; and 3H3, 7.5 μm (Fig. 1b), with less surface coverage and CR staining (Fig. 1a). We quantitated the CR mean relative fluorescent unit (RFU) values for the biofilms. Three of the four mAb-exposed samples had significantly reduced CR staining: 4G1 (23.9 RFU), 2C10 (22.1 RFU), and 3H3 (23.2 RFU), compared to the control and 6A samples (47.1 RFU and 48.1 RFU, respectively), although these were greater than the *csgBA* biofilm (7.91 RFU) (Fig. 1c).

As the mechanism of biofilm formation may differ depending on available surfaces, we also tested the effects of the mAbs on pellicle formation (e.g. a biofilm at an air–liquid interface). *S*. Typhimurium strains were cultured in 96-well plates for 3 days under biofilm promoting conditions, in the presence or absence of the human mAbs, stained with crystal violet, and examined by visual inspection. Pellicle biomass formation was robust in the control and 6A mAb cultures (Fig. 1d). In contrast, the samples exposed to 3H3 had virtually no pellicle, similar to the isogenic *csgBA* mutant, and the pellicles developed with the Aβ-specific mAbs were thinner than the controls (Fig. 1d). To quantitate the extent of pellicle disruption, we repeated these experiments and measured the total crystal violet binding to washed and dried pellicles (Fig. 1d). The optical densities were significantly reduced for wells treated with all four anti-Aβ mAbs. As 3H3 was the most effective of the mAbs evaluated, we investigated the affect of 3H3 on biofilm formation in several clinical isolates including *S. enteritidis*, *S. pullorum*, and uropathogenic *E. coli* UTI89 and *E. coli* MC4100, the common laboratory strain. Similar to our model organism, *S*. Typhimurium, 3H3 lead to a dose-dependent reduction in the pellicle biomass in *S. enteritidis*, *E. coli* UTI89, and *E. coli* MC4100 but not in *S. pullorum* (Supplementary Fig. 3). As the optical density for *S. pullorum* was overall lower compared to the other bacteria tested and 3H3 did not have an affect, it is possible that *S. pullorum* forms biofilms using bacterial factors than curli. Further experiments were conducted only with 3H3.

### 3H3 disrupts *S*. Typhimurium biofilm architecture and integrity.
The biofilm formed in the presence of the 3H3 mAb showed a diffuse Syto9 staining pattern (Fig. 1a). We hypothesized that this represented a change in the density of the biofilm, such that bacteria within the biofilm would be more distant from the inert

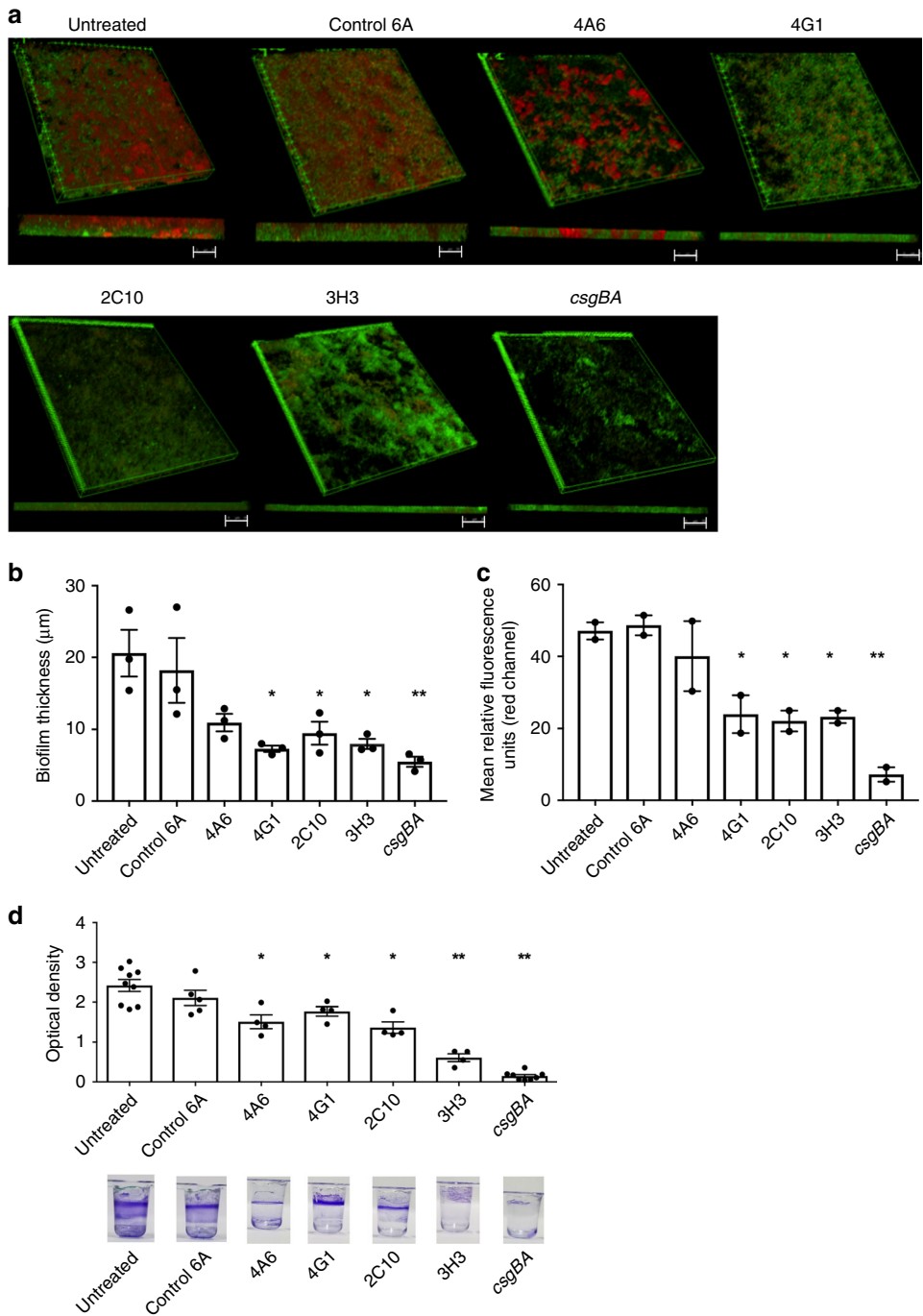

**Fig. 1 Incubation of *S.* Typhimurium biofilms with anti-amyloid mAbs reduces biofilm thickness and curli content. a** *S.* Typhimurium was cultured in the absence of mAb (untreated) or in the presence of 0.5 mg/ml control antibody 6A or 0.5 mg/ml 4A6, 4G1, 2C10, or 3H3. Isogenic mutant *S.* Typhimurium *csgBA* was included as a negative control. After 72 h, biofilms were stained with the bacterial stain Syto9 (green) and amyloid stain Congo Red (red), washed extensively, and imaged using a Leica TCS confocal microscopy at ×63. ImageJ was used to create 3D reconstructions of z-stacks using the 3D projection application. Scale bars represent 25 μm. **b** Biofilm thickness (μm) was determined from z-stacks using Leica TCS software. **c** Mean relative fluorescent units (RFU) of the red channel calculated from z-stacks using ImageJ. **d** Biofilms were grown in the absence (untreated) of antibody or in the presence of 0.5 mg/ml 6A, 4A6, 4G1, 2C10, or 3H3. *csgBA* was included as a negative control. After 72 h, biofilms were stained with crystal violet, and the optical density at 570 nm was determined. Representative images of crystal violet staining are shown below the graph. Mean and SE were calculated from results from at least two independent experiments. *$p < 0.05$, **$p < 0.01$ as determined by Student's *t*-test.

surface and less tightly adherent. Fresh *S.* Typhimurium biofilms were prepared on glass coverslips for 72 h, comparing cultures incubated with 6A, 3H3, or a polyclonal anti-CsgA serum that is known to completely inhibit biofilm formation. In these experiments, the biofilms were very gently washed prior to imaging to prevent any possible disruption of the biofilms. Using ImageJ

three-dimensional (3D) surface plot analysis, all particles within the biofilm were counted and pseudocolored green. This analysis revealed a difference in the topography of the biofilms (Fig. 2a). Curli-containing biofilms showed a dense topography of approximately 20 μm. However, the 3H3 and anti-CsgA exposed biofilms were less densely packed with an altered topography and

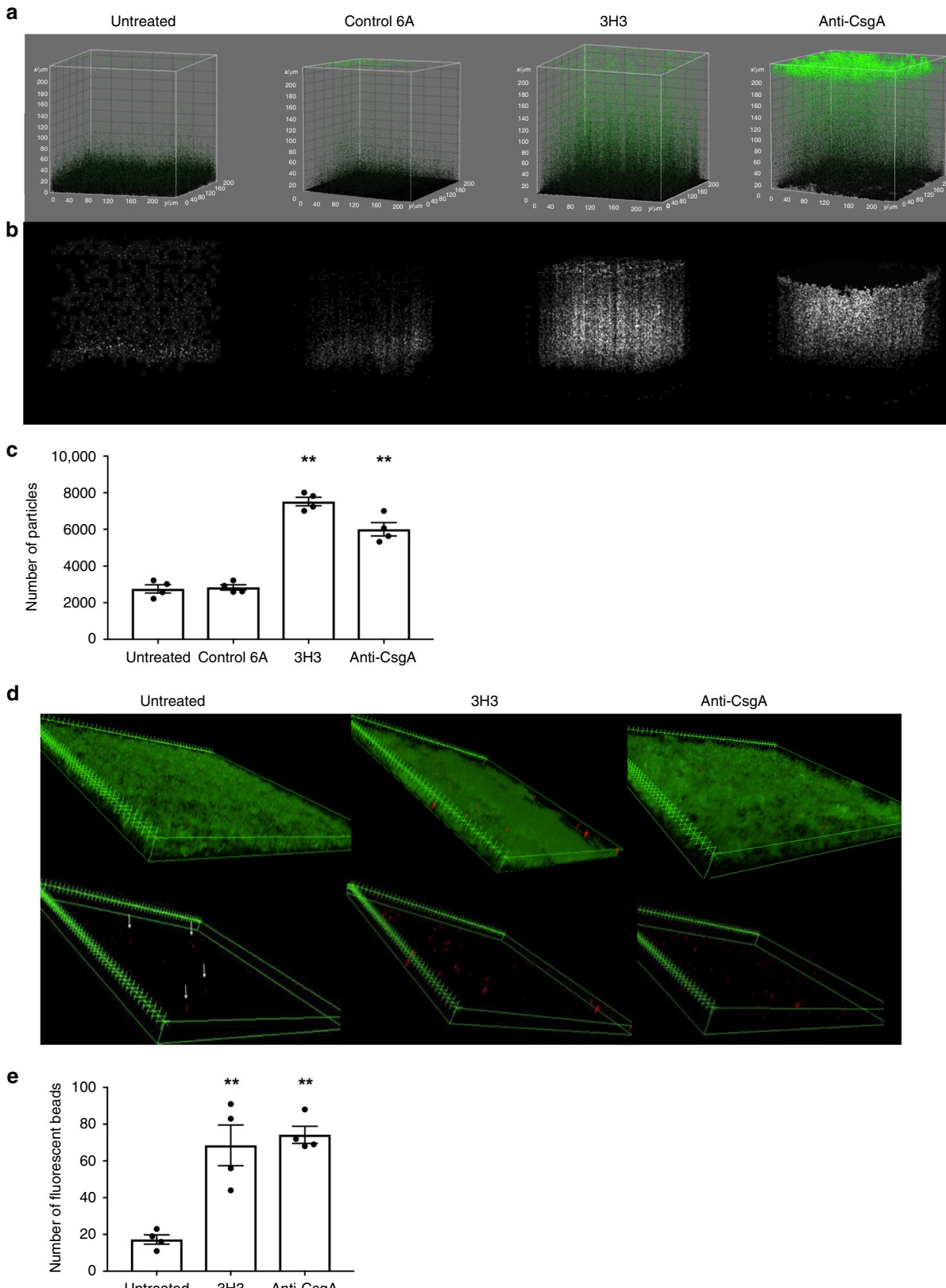

cells that appeared to be outside of the main biofilm mass (Fig. 2a). To quantify this population of cells, all particles above mean thickness of the untreated biofilm mass were identified and represented as white particles (Fig. 2b). The cells were counted using ImageJ (Fig. 2c). We next enumerated bacteria in the undisturbed supernatants by serial plating and found that biofilm disruption by 3H3 increased the release of bacteria (Supplementary Fig. 4).

These observations suggested the biofilm exposed to 3H3 may be more accessible to particles. We assessed the adherence and movement of 1 µm red-fluorescent glyoxylate beads applied to the top of the biofilms. We tracked the movement of the beads over

**Fig. 2 Incubation with 3H3 alters the biofilm architecture. a** *S.* Typhimurium biofilms were formed in the absence of mAb (untreated) or in the presence of 0.5 mg/ml 6A, 0.5 mg/ml 3H3, or anti-CsgA serum. After 72 h, biofilms were stained with Syto9 (green) and visualized using Leica TCS confocal microscopy at ×63. 3D surface plots were created in ImageJ and all Syto9 particles of the biofilms appear as green. Particles are colored black at the bottom of the z-plane and are increased in green intensity from 0 to 220 μm on the z-plane. **b** Particles above the mean biofilm mass of the untreated sample appear in white using the 3D surface plot application of ImageJ. **c** Number of particles above the mean biofilm mass (white particles) enumerated using ImageJ. **d** *S.* Typhimurium biofilms formed in the absence of mAb (untreated) or in the presence of 0.5 mg/ml 6A, 0.5 mg/ml 3H3, or anti-CsgA. After 72 h, biofilms were stained with Syto9 (green) and incubated with 10 μl Crimson FluoSpheres 1 μm red-fluorescent glyoxylate beads (red). Biofilms were imaged using confocal microscopy and biofilm projections of z-stacks were created using ImageJ (3D projections application). Scale bars represent 25 μm. Fluorescently labeled beads (red) were visualized by removing the Syto9 green channel. ImageJ was used to create 3D reconstructions of z-stacks. White arrows indicate location of beads in the untreated sample. **e** Number of red Crimson FluoSpheres within biofilms enumerated using ImageJ. Mean and SE were calculated from results of at least three independent experiments. *$p < 0.05$, **$p < 0.01$ as determined by Student's t-test.

20 min using time-lapse confocal microscopy (Fig. 2d, e, Supplementary Movies 1–3). In the control *S.* Typhimurium biofilm, vertical movement of the beads was observed, which was most likely caused by material properties of the biofilm. In biofilms formed with 3H3 or anti-CsgA, bead movement occurred both vertically and horizontally throughout the biofilm. Although equal amounts of beads were added to all samples, biofilms formed in the presence of 3H3 and anti-CsgA captured significantly more beads than the untreated controls. The average number of beads captured by the *S.* Typhimurium biofilms was 17, whereas the 3H3 and anti-CsgA exposed biofilms captured 68 and 75 beads, respectively (Fig. 2f). Taken together, these data suggest that inhibition of amyloid deposition prevents formation of a mature biofilm matrix, resulting in loose bacterial association and a porous structure.

**3H3 alters integrity of pre-established S. Typhimurium biofilms**. To determine if 3H3 can alter integrity of pre-established biofilms, biofilms of *S.* Typhimurium were established for 24 or 48 h. Biofilms were incubated for an additional 24 h in the presence or absence of 3H3 and then stained with Syto9, gently washed, and imaged using confocal microscopy. 3H3 exposure to biofilms established for 24 or 48 h inhibited further maturation of the biofilms and reduced total surface coverage (Fig. 3a). The 3H3 treated biofilms also displayed the characteristic unpacked topography (Fig. 3b). Visualizing the bacteria above the main biofilm mass as white particles allowed us to observe the incorporation of bacteria into the mature biofilm. 3H3 treatment reversed this process, even destabilizing the biofilm formed at the 48 h time point (Fig. 3c), and increasing the number of bacteria dissociating from the biofilm (Fig. 3d).

**3H3 inhibits the fibrillization of curli**. We previously observed that 3H3 inhibits polymerization of amyloid fibrils from Aβ and a lambda immunoglobulin associated with primary amyloidosis[38]. To test whether 3H3 affects curli fibrillization, we tested the synthetic peptides CsgA$_{R4-5}$ and CsgAR$_{4-5N122A}$ and the protein BSA in vitro using the Thioflavin T (ThT) assay[41]. CsgA$_{R4-5}$, which has the sequence of the fourth and fifth repeats of CsgA, self-associates and forms fibrils, whereas CsgAR$_{4-5N122A}$ contains a single amino acid substitution that prevents fibrillization[42]. Incubation of CsgA$_{R4-5}$ in the presence of 3H3 significantly reduced the polymerization of CsgA$_{R4-5}$ alone or incubated with the control antibody 6A (Supplementary Fig. 5A). We also evaluated the lag time ($t_0$) of fibrillization, the time required for monomers to self-associate and begin to fibrillize[43]. There was a significant increase in the lag time required for CsgA$_{R4-5}$ to self-associate when incubated with 3H3 in comparison to the calculated lag time for CsgAR$_{4-5}$ alone. Overall these data suggest that 3H3 inhibits initiation of CsgA fibrillization as well as elongation of curli fibrils (Supplementary Fig. 5B).

**3H3 and antibiotic treatment synergizes for biofilm eradication**. We reasoned that by altering the structure of the biofilm, 3H3 treatment would increase the susceptibility of the biofilm to antibiotics. We established biofilms of *S.* Typhimurium in the presence or absence of 3H3 for 48 h and then treated them with 30 μg/ml ampicillin for an additional 24 h. Biofilms were then stained with Syto9, washed extensively, and visualized using confocal microscopy. No alteration in overall biofilm appearance was observed when biofilms were treated with ampicillin alone (Fig. 4a), and the average thickness of biofilms treated with ampicillin alone did not differ significantly from that of untreated biofilms (16.2 vs 14.8 μm) (Fig. 4b). 3H3 treatment alone resulted in a biofilm with reduced surface coverage (10.07 μm) compared to the untreated sample (Fig. 4a, b). Treatment with 3H3 and ampicillin together resulted in a biofilm with very little surface coverage (Fig. 4a) and a biofilm thickness of 7.4 μm (Fig. 4b). In addition to ampicillin, we also tested two additional classes of antibiotics ciprofloxacin and streptomycin, belonging to the fluoroquinolone and aminoglycoside drug classes, respectively, in combination with 3H3. Using the same method as described for ampicillin, we incubated *S.* Typhimurium biofilm with ciprofloxacin at 0.125 μg/ml or streptomycin (12.5 μg/ml) alone or in combination with 3H3 and visualized the biofilms using confocal microscopy. Both antibiotics were ineffective on the biofilm alone (Supplementary Fig. 6A, B). However, when *S.* Typhimurium biofilms were incubated with 3H3 and ciprofloxacin together for an additional 24 h, there was a significant decrease in biofilm thickness and surface coverage in comparison to untreated or 3H3 treated biofilm (Supplemental Fig. 6A, B). Although not statistically significant, we observed a trend towards reduction in thickness and surface coverage in biofilms treated with 3H3 and streptomycin together (Supplemental Fig. 6A, B). Overall, these data show that 3H3-induced alterations to the biofilm extracellular matrix structure can facilitate antibiotic activity in vitro.

**3H3 promotes in vivo clearance of biofilms from pre-colonized catheters**. Gram-negative bacteria are a common cause of CLABSIs[13,44]. Biofilm-induced colonization of catheters leads to increased resistance to antibiotics and can necessitate catheter removal[20]. To investigate the effects of 3H3 on biofilm growth and architecture on catheters in vitro, we placed 1 cm sterile sections of an intravenous catheter in wells of a 24-well plate that contained biofilm-inducing media. We added 3H3 or left the wells untreated under these conditions and inoculated the cultures with *S.* Typhimurium. After incubation at 28 °C for 72 h, biofilms were stained with Syto9 and Congo Red and visualized using confocal microscopy. Untreated biofilms grew within the lumen and adhered tightly to the walls of the catheters. Biofilms grown in the presence of 3H3 had a loose architecture and did not contain curli (Fig. 5a).

As host innate immune responses are an integral factor in controlling bacterial biofilms, we investigated whether 3H3

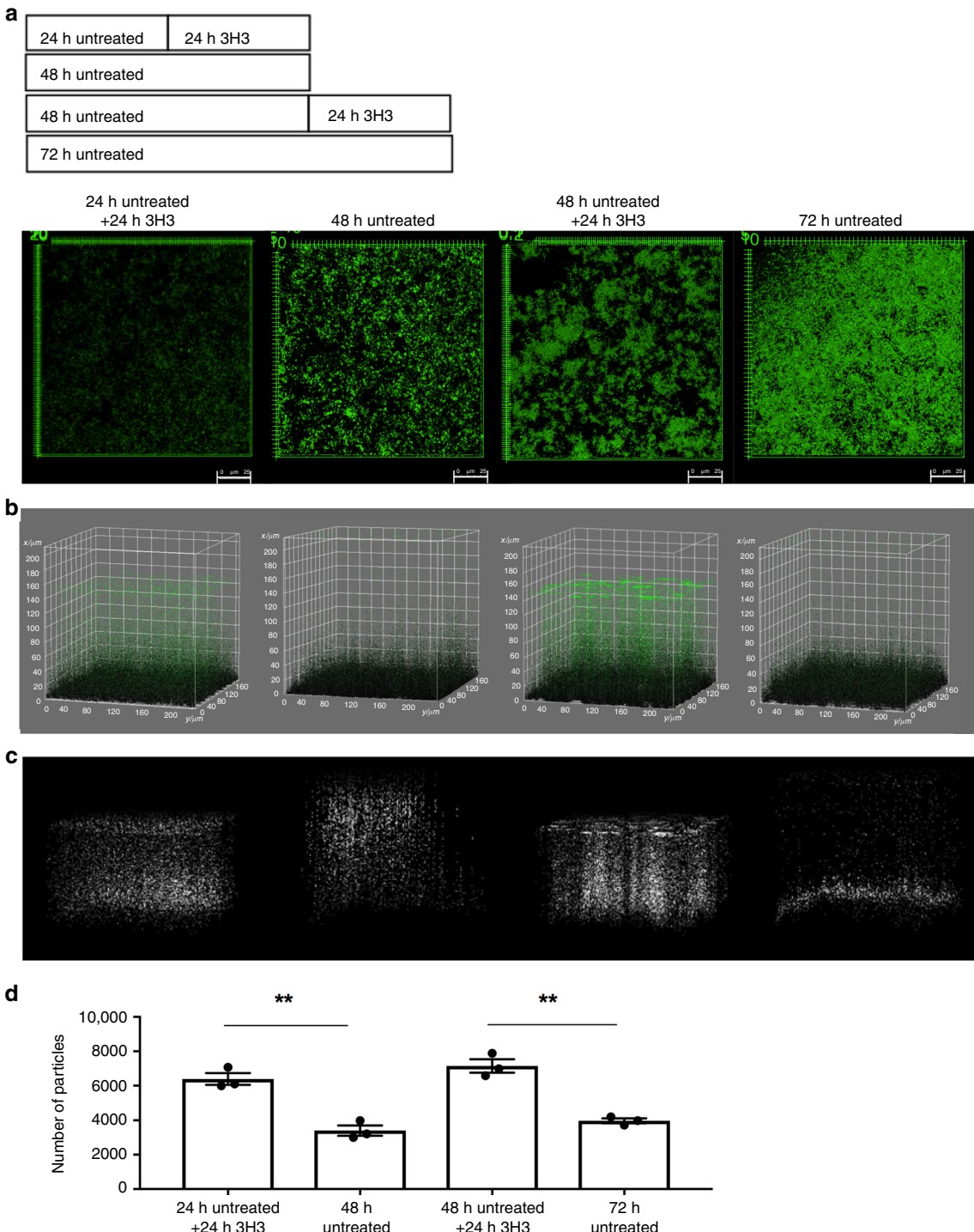

**Fig. 3 3H3 alters the architecture of pre-established *S.* Typhimurium biofilms. a** Experimental schematic and representative images of biofilms established for 24 and 48 h and then incubated for an additional 24 h without or with 0.5 mg/ml 3H3. Biofilms were stained with Syto9 (green) and visualized using Leica TCS confocal microscopy at ×63. Scale bars represent 25 μm. **b** 3D surface plots of biofilms depicted in **a** created in ImageJ. All biofilm particles are colored green. **c** 3D surface plots in which particles above the mean biofilm mass appear in white as determined by the 3D surface plot application in ImageJ. **d** Number of particles above the mean biofilm mass enumerated using ImageJ. Mean and SE were calculated from results from at least three independent experiments. *$p < 0.05$, **$p < 0.01$ as determined by Student's *t*-test.

treatment of catheters pre-colonized with *S.* Typhimurium would enhance biofilm clearance in vivo. We pre-colonized intravascular catheters for 24 h with *S.* Typhimurium and then implanted the catheters subcutaneously into the back flanks of BALB/c mice. At 24 and 48 h after catheter insertion, 100 μg 3H3 or buffer solution was percutaneously injected into the catheter lumens. At 72 h

after catheter insertion, mice were euthanized and the catheters were removed. Excised catheters were stained with Syto9 and Congo Red and visualized using confocal microscopy. On catheters from cohorts of mice not treated with mAb, biofilms formed a dense matrix with bacteria encapsulated within curli (Fig. 5b, upper left). In contrast, catheters from cohorts that

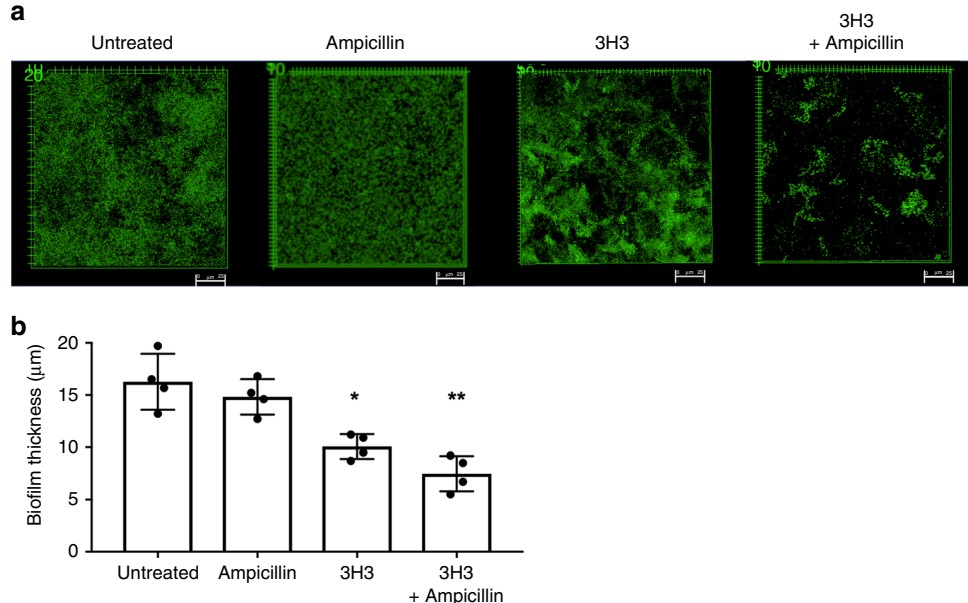

**Fig. 4 Synergistic effect of 3H3 and antibiotic reduces *S.* Typhimurium biofilm. a** *S.* Typhimurium biofilms were untreated or were incubated in the presence of 0.5 mg/ml control antibody 6A or 0.5 mg/ml 3H3. After 48 h, biofilms were subjected to 30 μg/ml ampicillin or not for an additional 24 h. Biofilms were stained with Syto9 (green), washed extensively, and visualized using Leica TCS confocal microscopy at ×63. Scale bars represent 25 μm. **b** Biofilm thickness (μm) was determined from z-stacks using Leica TCS software. Mean and SE were calculated from results from at least three independent experiments. *$p < 0.05$, **$p < 0.01$ as determined by Student's *t*-test.

received 3H3 had a loose biofilm architecture that was not tightly adhered to the catheters (Fig. 5b, upper right). To investigate the effect of a combination therapy of 3H3 with antibiotic on *S.* Typhimurium-colonized intravenous catheters in vivo, we treated mice with 1 mg/ml ampicillin in drinking water, starting 24 h prior to catheter insertion and continuing for the duration of the experiment. Catheters were injected percutaneously with 100 μg 3H3 or buffer solution 24 and 48 h after catheter insertion. At 72 h, catheters were removed and stained as above. Ampicillin treatment alone did not impact the architecture of the *S.* Typhimurium biofilms (Fig. 5b, lower left). In contrast, catheters treated with ampicillin and 3H3 in combination were virtually free of biofilm, with only minimal levels of Congo Red staining (Fig. 5b, lower right).

In the in vivo experiment, we observed round, Congo Red-positive structures, about the size of phagocytic immune cells, on the catheters treated with 3H3 (Fig. 5b upper right) suggesting that immune cells had phagocytosed the biofilm-associated bacteria. We tested whether 3H3 can trigger phagocytosis of biofilm-associated bacteria by innate immune cells using bone marrow-derived macrophages in vitro. We removed supernatants from untreated *S.* Typhimurium biofilms or from *S.* Typhimurium biofilms treated with 6A, 3H3, or anti-CsgA serum, and applied them to macrophage monolayers. After 1 h, we washed the macrophages to remove external bacteria and added gentamicin to kill any bacteria that had not been phagocytosed. Cells were then lysed to release the intracellular bacteria, and these bacteria were enumerated. Macrophages phagocytosed significantly more bacteria from the supernatants of the biofilm that had been incubated with 3H3 or anti-CsgA than from untreated supernatants or supernatants incubated with the 6A control mAb (Fig. 6a). To account for possible differences in the numbers of bacteria in the supernatants, the supernatant optical densities were adjusted to 0.5 prior to the addition to macrophages. Again, more bacteria were recovered from the macrophages that were stimulated with the supernatants of the biofilm that had been incubated with 3H3 or anti-CsgA (Fig. 6b).

To investigate if 3H3 enhances the uptake of curli into immune cells, we added Congo Red-labeled curli fibers incubated with or without 3H3 for 1 h to a culture of immortalized macrophages. After 1 h, cells were washed to remove excess curli. Curli internalization by the macrophages was visualized using confocal microscopy. More curli was detected inside the macrophages when it had been pre-incubated with 3H3 (Fig. 6c). These data suggest that 3H3 disrupts biofilm structure and enhances macrophage uptake of bacteria and curli fibers, which may promote clearance of biofilms by the innate immune system and reduce bacterial re-seeding.

## Discussion

Curli fibrils are the major component of *Enterobacteriaceae* biofilms, which protect the bacteria from antibiotics and immune cells[28,35,45]. Curli fibrils are amyloids, insoluble polymers that have a characteristic β-sheet structure[30,46,47]. Inhibiting curli polymerization has been proposed as a general model for treating *Enterobacteriaceae* biofilms[48]. Amyloid proteins with highly divergent sequences can share conformational epitopes that are recognizable by mAbs[49,50]. Some mAbs that bind pan-amyloid epitopes inhibit fibrillization of eukaryotic amyloids and have anti-amyloid activities in vivo[38,49,51,52], but this has not been shown for prokaryotic amyloids.

Each of the four amyloid-binding mAbs tested here disrupted biofilm formation in vitro. 3H3 had been extensively characterized in previous studies[38] and proved most effective. 3H3 is a human mAb specific for a pan-amyloid epitope that is present on a wide variety of pathogenic human amyloids, including Aβ, immunoglobulin light chain, transthyretin, and tau[38]. 3H3 inhibits amyloid fibril deposition in vitro and in animal models of AD and familial Danish dementia. Here, we describe how 3H3 also binds the bacterial amyloid curli and inhibits polymerization of curli monomers, which correlates with profound effects on the structure and function of the biofilm in vitro and in vivo. 3H3 mAb antagonized *S.* Typhimurium biofilm formation on glass

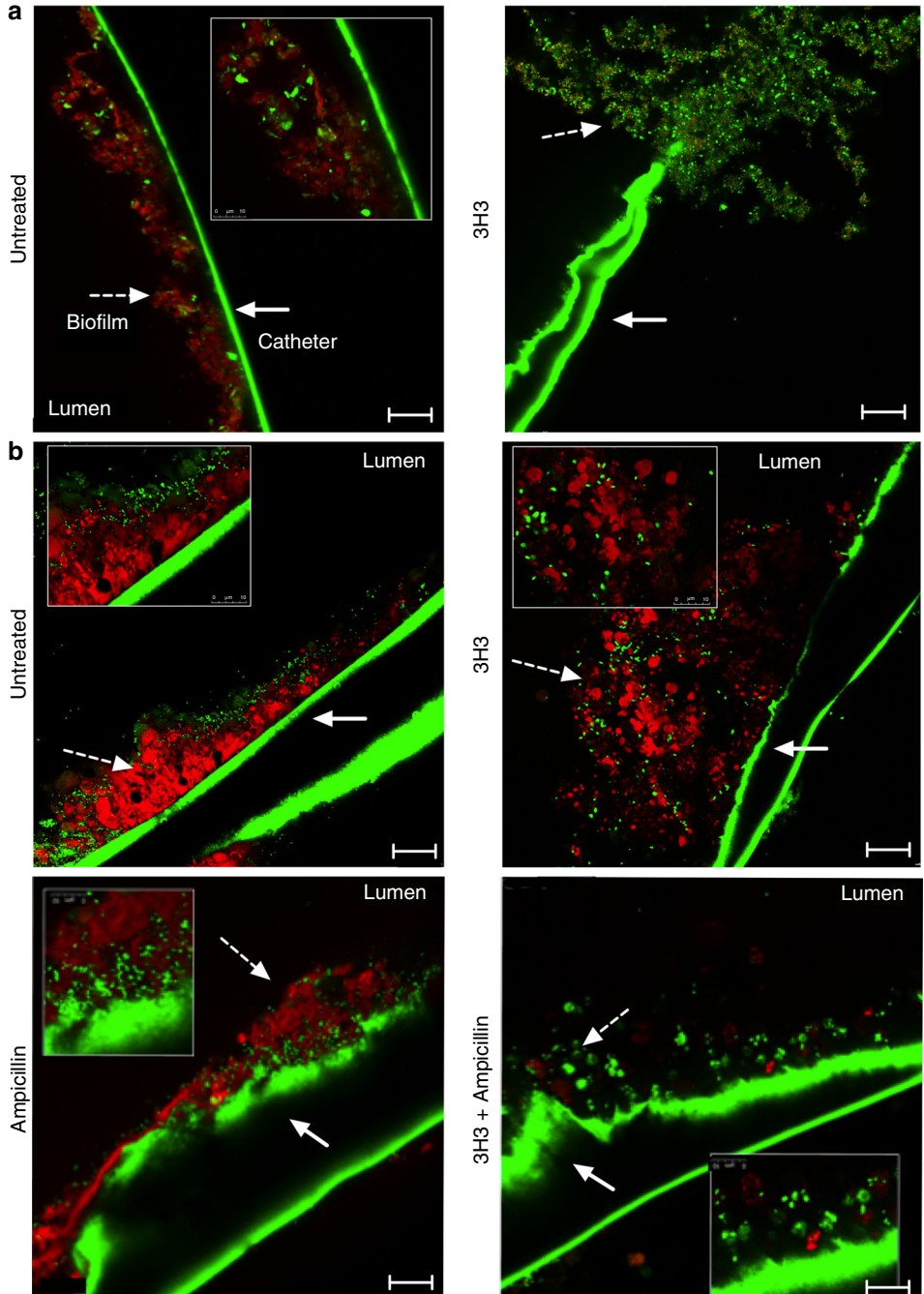

**Fig. 5 Combination treatment of 3H3 and antibiotic leads to biofilm eradication. a** Biofilms (dashed white arrow) were established in the presence (untreated) or absence of 0.5 mg/ml 3H3 on medical grade catheters in vitro. After 72 h biofilms were stained with Syto9 (green) and amyloid curli stain Congo Red (red) and imaged using Leica TCS confocal microscopy at ×63. Scale bars represent 25 μm. Inserts are ×3 zoom images. Catheters exhibit green autofluorescence (solid white arrow). **b** Biofilms were established on medical grade catheters in vitro 24 h prior to insertion of catheters into flanks of mice. At 24 and 48 h post insertion, 100 μg 3H3 was injected in vivo percutaneously into the catheter lumen. Where applicable, drinking water was supplemented with 1 mg/ml ampicillin beginning 24 h prior to catheter insertion. At 72 h after catheter insertion mice were euthanized and catheters were removed, stained with Syto9 (green) and Congo Red (red), and imaged using a Leica TCS confocal microscopy at ×63. Scale bars represent 25 μm. Insets are ×3 images. Catheters exhibit green autofluorescence.

slides, at the air–liquid interface (pellicles), and on vascular catheters. The extracellular matrix of the 3H3-treated biofilms failed to fully mature into an impermeable barrier with tightly embedded bacteria. Rather, we observed a "loose" topography, in which bacteria remained relatively distant from the surface and could be removed by washing. 3H3 also disrupted the structure of established *S.* Typhimurium biofilms. Treatment with 3H3

decompacted a 48-h biofilm, allowing bacteria to move away from the surface. The decrease in density of the biofilm exposed to 3H3 was demonstrated by increased in penetration of 1 μm glyoxylate beads in the biofilm. The 3H3-treated biofilm displayed changes in the extracellular matrix composition and structure, upon loss of curli incorporation which facilitated lateral movement of the glyoxylate beads within the biofilm. Finally, 3H3 enhanced the

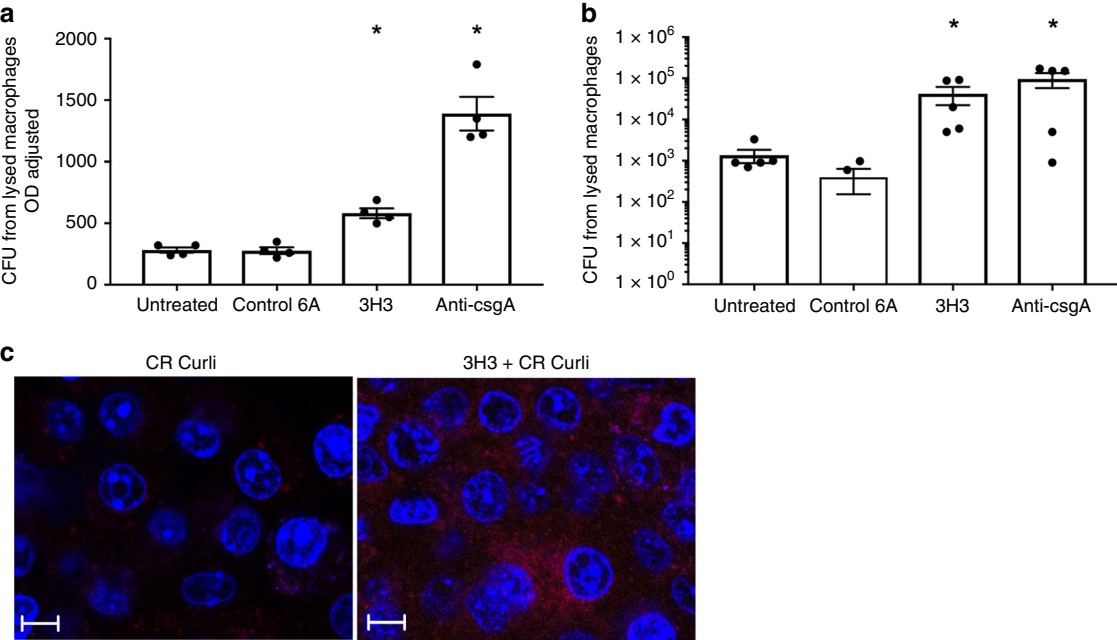

**Fig. 6 Treatment with monoclonal antibody 3H3 enhances uptake of bacteria by macrophages. a** Wild-type bone marrow derived macrophages (BMDMs) were treated with 100 µl supernatants from biofilms grown in the absence (untreated) or presence of 0.5 mg/ml 3H3. After 1 h, gentamicin was added, cells were lysed, and bacteria were enumerated as colony forming units. **b** Wild-type BMDMs were treated with supernatants from biofilms grown in the absence (untreated) or presence of 0.5 mg/ml 3H3 adjusted to an optical density of 0.5 at 600. After 1 h, gentamicin was added, cells were lysed, and bacteria were enumerated as colony forming units. **c** $1 \times 10^6$ wild-type immortalized macrophages were stimulated with 5 µg of untreated curli (CR Curli) or 3H3 incubated curli (3H3 CR Curli) for 1 h at 37 C and 5% $CO_2$. Cells were washed 3× with PBS to remove excess curli and macrophages were stained with DAPI (blue). Internalization of the curli (red) was imaged using a Leica TCS confocal microscopy at ×63. Scale bars represent 25 µm. Mean and SE were calculated from results from at least three independent experiments. *$p < 0.05$, **$p < 0.01$ as determined by Student's $t$-test.

activity of ampicillin against biofilms formed on a glass slide and on a vascular catheter. Importantly, 3H3 collaborated with ampicillin to clear a biofilm from a vascular catheter implanted subcutaneously in mice. These data further support the concept of inhibiting curli polymerization as an anti-biofilm strategy and establish the therapeutic potential of the 3H3 mAb to treat curli-containing biofilms.

Curli is secreted as a monomer (CsgA) that self-associates into small oligomeric subunits in the extracellular matrix. Small oligomeric subunits elongate though the addition of CsgA until the mature curli fibril is formed[28,31,36]. Curli amyloid shares structural and functional features with non-bacterial amyloids. Atomic resolution imaging revealed that curli amyloid fibrils resemble the Aβ fibrils found in Alzheimer's disease[53], and synthetic D-enantiomeric peptides designed to inhibit Aβ fibrillization also inhibit CsgA fibrillation. Curli fibrillization is also inhibited by human transthyretin, an amyloid protein responsible for cardiomyopathy and neuropathy[54]. These findings suggest the existence of pan-amyloid epitopes that are required for amyloid fibrillization and that polypeptides that bind these epitopes can be used to interrupt the maturation of amyloid structures.

Upon incubation of synthetic CsgA peptides with 3H3, we observed a reduction in curli fibrillization. 3H3 and 4G1 had the most activity against the *S*. Typhimurium biofilms, which correlated with relatively high binding affinity to Aβ oligomers, in contrast to 4A6 and 2C10, which had a strong preference for Aβ fibrils (Supplementary Table 1, Supplementary Fig. 2). In addition, 3H3 and 4G1 mAbs inhibited both curli and Aβ fibrillization, whereas 4A6 did not (Supplementary Fig. 2). The activity of 3H3 against curli is consistent with what we previously observed with Aβ and suggests that the mAb binds a functional epitope on CsgA that is present on many human amyloids, even though its

primary amino acid sequence differs greatly from the human proteins. Since the 3H3 was cloned from a healthy individual, it may be that the antibody originated as part of his immune response to a bacterial amyloid. Together, 3H3 and 4G1 define a category of mAbs that (a) bind pan-amyloid epitopes shared by eukaryotic and prokaryotic proteins and (b) inhibit amyloid fibrillization. Antibodies with these capabilities have utility for treating many different amyloid diseases.

Despite their functional similarities, 3H3 and 4G1 were cloned from different individuals and have highly divergent variable domain sequences with evidence of substantial affinity maturation (data not shown). These observations suggest that the curli amyloid polymerization domain contains multiple pan-amyloid epitopes that could be targeted by combinations of antibodies with complementary activities, potentially replicating the potent anti-fibrillar activity of the anti-CsgA antiserum. In addition, 3H3 and 4G1 variants (or new mAbs) may be discoverable that recognize CsgA-specific amino acids simultaneously with the pan-amyloid epitope. Such mAbs may be more potent anti-amyloids because of higher affinity, increased steric hindrance, or allosteric effects. Lastly, it is possible that pan-amyloid-binding activity is not solely required for mAbs to inhibit curli polymerization. Instead, it may be that pan-amyloid epitopes define a particular region of curli important for oligomerization. An ideal mAb combination could target both pan-amyloid and curli-specific epitopes in this region.

The ability of 3H3 to inhibit polymerization of curli monomers suggests that some of its anti-biofilm activity may involve inhibiting the development of full-length curli fibrils. The observation that 3H3 binds and inhibits polymerization of amyloids with dramatically different primary sequences suggests that it binds a pre-fibrillar intermediate conformation that is shared by diverse

amyloids. In the case of curli, these structures may consist of curli monomers or low molecular weight oligomers[32]. Maintenance of curli in an un-polymerized state could inhibit its incorporation into the extracellular matrix and would be consistent with the loose topography observed. However, 3H3 also demonstrated activity against established biofilms in vitro, loosening the biofilm topography and allowing bacteria to migrate away from the surface of the slide (Fig. 3). As curli is important for binding to abiotic surfaces[27,55,56], it is possible that 3H3 interferes with the domains of curli required for surface binding.

Antibiotics that belong to the beta-lactam, fluoroquinolone, and aminoglycoside drug classes all demonstrated increased cell killing when the biofilm extracellular matrix was altered by 3H3. These experiments used previously determined MICs applicable to short-term exposures. However, the effects of biofilm disruption on antibiotic sensitivity will likely depend on whether the mechanism of cell killing is concentration dependent (fluoroquinolones, aminoglycosides) or time-dependent (beta-lactams). Additional studies will be necessary to explore these interactions in vitro and in vivo. Clearance of curli and curli-associated bacterial cells by phagocytic cells may have contributed to eradication of the vascular catheter biofilm in vivo. Antibodies have been shown to mediate macrophage-induced clearance of eukaryotic amyloids in vivo[57–59]. The biofilm exposed to 3H3 in vivo replicated the loose architecture seen in vitro but was also notable for Congo Red stained structures that were approximately the estimated size of phagocytic cells. Macrophage experiments showed that the 3H3-treated biofilms release bacteria in a form that can be phagocytosed by macrophages, and that uptake is enhanced by 3H3 binding to the bacteria/curli complexes.

A therapeutic strategy against biofilms needs to clear device-associated biofilms while reducing the development and spread of antibiotic resistance. For these purposes, a pan-amyloid mAb such as 3H3 offers many potential benefits. It could be given in combination with antibiotics to raise their effective concentrations sufficiently to clear infected medical devices. It could also be used as a device coating to prevent new biofilm formation. Bacteria should not develop resistance to pan-amyloid mAbs, as they would not affect the growth of planktonic bacteria. Furthermore, their ability to opsonize biofilm bacteria may reduce the systemic seeding that can occur when a biofilm is dissociated. Human mAb therapeutics have a strong safety record in general, and because the 3H3 was cloned from a healthy individual and binds a non-native antigen, it would be predicted to have few off-target effects. Human mAbs have half-lives of 2 weeks or more, which may be ideal for prevention and treatment of infections associated with curli biofilms. Lastly, as amyloids are widespread and found in biofilms of bacteria belonging to various phyla (Bacteroidetes, Proteobacteria, Firmicutes, and Thermodesulfobacteria)[60,28] including important human pathogens such as *Staphylococcus aureus*[47], our findings may be generalizable to the treatment of many different bacterial biofilms.

## Methods
**Bacterial strains and growth conditions**. *S.* Typhimurium strain IR715 is a fully virulent, nalidixic acid-resistant strain derived from the ATCC strain 14028 (ref. [61]). *S.* Typhimurium IR715 *csgBA*[40] and *S.* Typhimurium IR715 *msbB* were previously described[62]. Bacteria were grown in Luria Bertani (LB) broth supplemented with 50 μg/ml nalidixic acid when appropriate. Clinical isolates of *S. enteritidis* and *S. pullorum* (ATCC 9120, isolated from a patient with diarrhea) were kindly provided by Dr. Andreas Baumler at UC Davis. Uropathogenic *E. coli* UTI89 (isolated from a patient with an acute bladder infection) and *E. coli* MC4100 were kindly provided by Dr. Scott Hultgren from Washington University in St. Louis. To induce biofilm formation, overnight bacterial cultures of *Salmonella* species were diluted 1:100 in LB No Salt medium and *E. coli* species were diluted 1:100 in LB low salt. All biofilm cultures were grown statically for 72 h at 28 °C. YESCA agar was prepared as described previously[33].

**Isolation of antibodies**. The 3H3 mAb has been previously described[38]. Additional mAbs reactive with oligomeric Aβ were cloned from B cells from an elderly female with a clinical diagnosis mild-moderate Alzheimer's disease (Supplementary Methods). The sample was collected under a protocol approved by the Main Line Hospitals Institutional Review Board and was consistent with the principles set out in the WMA Declaration of Helsinki and the NIH Belmont Report. The isotype control human mAb 6A (anti-botulinum toxin antibody) was described previously[39]. Anti-CsgA serum was obtained from rabbits as previously described[63].

**Biofilm assays**. The crystal violet assay was performed as described previously[64]. To investigate the curli content and thickness of biofilms grown in the presence or absence of antibody using confocal microscopy, overnight cultures of wild-type *S.* Typhimurium and the isogenic *csgBA* mutant were diluted 1:100 in LB No Salt broth with 0.5 mg/ml of 3H3, 4G1, 4A6, 2C10, or anti-CsgA. Biofilm formation was analyzed at 24, 48, and 72 h. Biofilms were washed four times to remove planktonic bacteria and then stained with 3 μl/ml Syto9 (Invitrogen) for 15 min and then washed four times with phosphate-buffered saline (PBS). Biofilms were then stained with 10 μg/ml Congo Red (Sigma Aldrich) for 15 min and washed four times with PBS. Congo Red was visualized at an excitation of 561 nm and an emission of 650–750 nm and Syto9 was visualized at an excitation of 483 nm and an emission of 503 nm at ×63 using a Leica SP5 Microscope equipped with a TCS confocal system. To image the overall architecture of the biofilms, the same protocol was used but the mature biofilms were only washed once with PBS and Congo Red was not applied. 3D surface plots were created using ImageJ Software.

To investigate the increasing concentrations of 3H3 on biofilms bacteria were diluted 1:100 in LB No Salt broth with 0.1, 1, 10, 25, 50, 250, and 500 μg/ml 3H3 for 72 h statically at 28 °C. Crystal violet assay was performed.

**Bacterial enumeration from biofilm supernatants**. To enumerate bacteria present in supernatant due to altered biofilm architecture, 100 μl supernatant was collected. Ten-fold serial dilutions were plated on LB agar plates with appropriate selection to determine the number of colony-forming units.

**Culturing of macrophages**. Immortalized macrophage cells (IMMs) derived from wild-type mice (NR-9456) were obtained from BEI Resources. IMMs were maintained in DMEM (Invitrogen) supplemented with 10% heat-inactivated FBS (Life Technologies) and were grown in a humidified incubator at 37 °C with 5% $CO_2$.

Bone marrow-derived macrophages (BMDMs) were generated from 6- to 8-week-old female wild-type C57BL/6 mice (Jackson Laboratories). Macrophages were isolated and differentiated as described previously[65]. The differentiated BMDMs were plated at $5 \times 10^5$ cells per well in a 48-well plate and allowed to adhere for 24 h in a humidified incubator (37 °C in 5% $CO_2$) prior to experimentation.

**Purification of curli amyloid and Congo Red labeling**. The purification of amyloid curli has been previously described[66]. The concentration of the curli fibers was determined using BCA reagent (Novagen) according to the manufacturer's protocol. Purified curli was labeled with Congo Red as described previously[67].

**Macrophage uptake assays**. Overnight cultures of *S.* Typhimurium were diluted 1:100 in LB No Salt broth in the presence or absence of 0.5 mg/ml mAb and grown statically at 28 °C. After 72 h, 100 μl culture supernatant was collected and stained with 3 μl/ml Syto9. An additional aliquot of 100 μl was added to BMDM monolayers in 24-well plates. Macrophages and bacteria were incubated in a humidified chamber at 37 °C in 5% $CO_2$. After 1 h, 10 μg/ml gentamicin (Fisher) was added, and samples were incubated for 1 h. BMDMs were washed three times with PBS and then lysed with 0.01% Triton X-100 for 30 min in a humidified chamber at 37 °C in 5% $CO_2$. To enumerate the bacteria within the BMDMs, 1:10 serial dilutions of the BMDM lysate were collected, and the numbers of colony-forming units were determined by plating on agar plates. To visualize bacteria uptake, the BMDMs were first stained with 15 μM CellTracker Blue (Molecular Probes) for 15 min according to the manufacturer's protocol and processed as above. Bacteria were stained with Syto9 and analyzed as described above.

To determine curli uptake, Congo Red-labeled curli (100 μg) was incubated with or without 0.5 mg/ml 3H3 at room temperature with gentle shaking. After 1 h, $5 \times 10^5$ wild-type IMMs were stimulated with 10 μl/ml Congo Red-labeled curli with or without 3H3. After 1 h at 37 °C with 5% $CO_2$, cells were washed three times with sterile PBS and then incubated with 0.01% Triton X-100 for 30 min at 37 °C in 5% $CO_2$. Cells were transferred to wells of a clear bottom, black, 96-well microplate. Fluorescence of Congo Red-labeled curli was measured using a Flex Station (Molecular Devices) at an excitation of 497 nm and emission at 614 nm using a top read.

To visualize the uptake of Congo Red-labeled curli by IMMs using confocal microscopy, Congo Red-labeled curli was incubated with or without 3H3. One day prior to stimulation, sterile circular glass coverslips (Fisher) were inserted into wells of sterile 24-well tissue culture dishes, and each well was seeded with $1 \times 10^6$ wild-type IMMs. After the macrophages had adhered to the glass coverslip, cells were treated with 5 μg/ml Congo Red-labeled curli previously incubated with or without 0.5 mg/ml 3H3 and 1 μg/ml DAPI (Invitrogen). After 30 min, cells were washed

three times with sterile PBS. Coverslips were placed on slides with 3 μl Vectashield (H-100). Cells were imaged using confocal microscopy at ×63. Congo Red was visualized at an excitation of 497 nm and emission at 614 nm, and DAPI was visualized at an excitation of 358 nm and an emission of 461 nm.

**Evaluation of biofilm topology with a bead assay**. Pellicle biofilms of *S.* Typhimurium were cultured on glass coverslips and stained with Syto9 as described above. After biofilms established, 10 μl fluorescently labeled beads (Crimson FluoSpheres, 1.0 μm, Life Technologies) were added on top of the biofilm. After a 10-min incubation at room temperature, excess beads were removed by washing the biofilm gently with sterile PBS three times. Coverslips were removed from the dishes and placed upside down in wells of an eight-well Multi-test slide (MP Biomedicals) with 3 μl Vectashield (H-1000) and sealed with fast drying clear nail polish (Fisher). Time-lapse bead movement was visualized using a Leica SP5 Microscope with a TCS confocal system taken at ×63. Syto9 was visualized at an excitation of 483 nm and an emission of 503 nm. Crimson FluoSpheres were visualized at an excitation of 625 nm and an emission of 645 nm.

**Antibiotic and mAb combination treatment of biofilms**. To determine the effect of a combination of mAb and antibiotic treatment on biofilm formation, biofilm growth was induced in the presence or absence of 0.5 mg/ml 3H3, control 6A, or anti-CsgA. After 48 h, 30 μg/ml ampicillin was added. After an additional 24 h, biofilms were washed with PBS and stained with Syto9 and imaged using confocal microscopy as described above. Overnight cultures of wild-type *S.* Typhimurium were diluted 1:100 in LB No Salt broth with or without 0.5 mg/ml of 3H3 and grown for 72 h at 28 °C. After 72 h, 0.125 μg/ml ciprofloxacin or 12.5 μg/ml streptomycin was added to appropriate biofilms for an additional 24 h. Biofilms were washed four times to remove planktonic bacteria and then stained with 3 μl/ml Syto9 (Invitrogen) for 15 min and then washed four times with PBS. Syto9 was visualized at an excitation of 483 nm and emission of 503 nm at ×63 using a Leica SP5 Microscope equipped with a TCS confocal system. 3D surface plots were created using ImageJ Software.

**Murine catheter implantation model**. Prior to implantation, 2-mm sterile catheter pieces (Hospira Sapphire, AP431-01) were pre-colonized for 24 h at 28 °C with an overnight culture of *S.* Typhimurium diluted 1:100 in LB No Salt Broth. The catheters were then inserted through percutaneous tunnel incisions into the backs of six- to eight-week-old female BALB/c mice (Jackson Laboratory) using sterile techniques under isoflurane anesthesia. Mice were anesthetized with iso-flurane at 24 and 48 h after catheter insertion and 100 μg 3H3 was percutaneously injected into the catheter lumens. Some of the mice received 1 mg/ml of ampicillin (Sigma) mixed into water given ad libitum, starting 24 h prior to catheter insertion. Mice were euthanized 72 h after catheter implantation. Excised catheters were stained with Syto9 and Congo Red and visualized as described previously[67].

**Aβ monomers, ADDLs, oligomers, and fibrils**. Synthetic Aβ 42 monomers biotin-labeled at either the N or C terminus were obtained from Anaspec (Fremont, CA). Aβ globular oligomers were prepared as previously described[68]. Briefly, Aβ 42 monomer (Life Technologies, Grand Island, NY) was dissolved at 5 mM in dry DMSO, sonicated in a bath sonicator for 30 min, adjusted to 400 μM with PBS, following the addition of 1/10th volume of 2% SDS, the mixture was incubated at 37 °C for 6 h. After dilution with three volumes of water, the mixture was incubated for an additional 18 h at 37 °C and centrifuged at $3000 \times g$ for 20 min at RT to remove any Aβ1–42 fibrils. The supernatant was then concentrated using a Vivaspin centrifugal concentrator with a 30 kDa molecular weight cut-off (Sar-torius Stedim, Bohemia, NY), dialyzed against PBS, characterized by SDS:PAGE, and stored at 4 °C for up to 2 weeks. To create stable SPR sensors, oligomers were cross-linked by treatment with 1 mM glutaraldehyde for 15 min at RT followed by adjusting to 5 mM ethanolamine. The ethanolamine was removed by ultrafiltration using a 10 kDa cut-ff centrifugal concentrator. Aβ fibrils were prepared by a modification of the method of O'Nuallain and Wetzel[50]. Hexafluoro-2-propanol (HFIP)-treated Aβ monomer was dissolved in freshly diluted 2 mM NaOH. After gentle agitation, the solution was centrifuged at $10,000 \times g$ for 60 min to remove large clumps or fibrils. The supernatant was then adjusted to 1 × PBS, 0.05% sodium azide, and incubated with agitation at 37 °C for 14 days. Tau PHF were isolated from brain tissue obtained from AD patients at autopsy using the method of Greenberg and Davies[69].

**Human monoclonal antibody cloning**. Human mAbs reactive with Aβ ADDLs were obtained from a blood sample of a female in her 70s (Patient ID: AD13_7) with a clinical diagnosis of mild to moderate Alzheimer's disease. The sample was collected under a protocol approved by the Main Line Hospitals Institutional Review Board and was consistent with the principles set out in the WMA Declaration of Helsinki and the NIH Belmont Report. Following receipt of informed consent, blood was collected, and CD27+ mononuclear cells were iso-lated and fused to the B5-6T fusion partner cell line following the published methods[70]. Following HAT selection, hybridoma supernatants were screened by ELISA for secretion of IgGs that bound to ADDLs Aβ1–42 oligomers and antigen-

specific IgGs were detected with the HRP-conjugated anti-human IgG Fc fragment-specific monoclonal antibody (9040-05, Southern Biotechnology). Stable, antigen-specific mAbs (including 4G1, 4A6, and 2C10) were isolated by three rounds of cloning in semi-solid medium. For scale-up, hybridomas were adapted to advanced RPMI with 5% ultra-low IgG medium (Life Technologies) and incubated for 5 days in a 500-ml roller bottle. Filtered supernatants were purified over protein G-Sepharose (GE Healthcare Life Sciences). Antibody concentrations were deter-mined using a NanoDrop Spectrophotometer (NanoDrop Technologies, Wil-mington, DE). The isotype control human mAb 6A (anti-botulinum toxin antibody) was described previously[39]. IgG heavy chain and light chain subtypes were determined by ELISA, and variable domain DNA sequences were obtained as previously described. Sequences were analyzed with the IMGT website (http://www.imgt.org)[71].

**Biotinylation of sensor ligands**. Aβ globular oligomers (ADDLs) were biotin labeled with NHS-PEO$_{12}$-Biotin (Pierce, Rockford, IL) as specified by the manu-facturer. The molar excess of biotinylation reagent to Aβ oligomers was between 20- to 60-fold; higher ratios were used when the concentration of oligomer was less than 2 mg/ml. After 2 h at 4 °C, excess biotin reagent was removed by dialysis against PBS. IAPP fibrils, Aβ fibrils, and tau PHF were biotinylated using this same procedure.

**SPR measurements**. Biotinylated Aβ monomer, oligomer, fibril, tau PHF, and IAPP preparations were each immobilized on the gold surfaces of SensiQ CO$_2$H sensors using a neutravidin-biotinylated target molecule immobilization protocol developed by the manufacturer (SensiQ Technologies, Oklahoma City, OK). Briefly, the sample channel was activated for conventional amine coupling by passing 50 μl of EDC/NHS activation solution over the channel at 25 μl/min. This was followed by a 100 μL injection of 100 μg/ml solution of Neutravidin, in 10 mM acetate buffer pH 4.5, over the activated surface for 20 min at 5 μl/min. This was followed by 100 μL Biotin tagged Aβ (50 μg/ml) injected over the Neutravidin-coated surface at 25 μl/min. After 1 h equilibration, the sensor was prepared for binding studies with 10 mM phosphoric acid.

For binding assays, 100 μl mAb solution was co-injected into the SPR apparatus at a flow rate of 50 μl/min. Five minutes after the injection was completed, the sensor surfaces were regenerated with 100 μl of 10 mM phosphoric acid at a flow rate of 50 μl/min. The binding curves, disassociation constants ($K_d$), maximum response ($R_{max}$), and their respective standard deviations were determined using single component fits using Qdat Data Analysis Software (SensiQ Technologies).

**Inhibition of fibril formation**. We used dynamic light scattering to measure amyloid fibril formation in solution. $Z_{ave}$ measurements of 0.25 ml of Aβ globu-lomers (50 μg/ml) in PBS pH 7.4 were made without or with the addition of 2 μg human mAb. The $Z_{ave}$ measurements were taken for an hour using a Zetasizer Dynamic Light Scattering apparatus (Malvern Panalytical, Malvern UK).

**Thioflavin T fibrillization assay**. Synthetic peptides that correspond to the fourth and fifth repeats of CsgA, CsgA$_{R4-5}$, and CsgA$_{R5-4N122A}$ described previously[42] were synthesized by Biosynthesis Inc. Polymerization of the synthetic peptides was previously described[43]. To monitor the fibrillization of the synthetic CsgA peptide, 100 μM CsgA$_{R4-5}$, or 100 μM CsgA$_{R5-4N122A}$ was mixed with an equal volume 10 μM ThT in the presence or absence of 0.5 mg/ml mAb 3H3 or control antibody 6A in a black Nunc 96-well plate (Fisher). The plate was sealed with an optical plate cover (Eppendorf). Fluorescence of ThT (excitation 440 nm/emission 490 nm) was monitored at 37 °C using a BMG Labtech POLARstar Omega plater reader. Readings were taken at 8-min intervals for 36 h. Lag time ($t_0$) was calculated using the following formula: $t_0 = t_{1/2} - 2t_e$, where $t_{1/2}$ is the time required to reach half the maximum fluorescence intensity and $t_e$ is the elongation time[43].

**Ethics statement**. All animal experiments were performed in AALAC-accredited animal care and use program at the Lewis Katz School of Medicine, with protocol (#4711) approved by the Temple University Institutional Animal Care and Use Committee in accordance with guidelines set forth by the USDA and PHS Policy on Humane Care and Use of Laboratory Animals under the guidance of the Office of Laboratory Animal Welfare (OLAW).

**Statistical analysis**. Data were analyzed using GraphPad Prism software. Two-tailed Student's t-tests were used as appropriate. The p values < 0.05 were con-sidered significant.

**Reporting summary**. Further information on research design is available in the Nature Research Reporting Summary linked to this article.

## Data availability

Source data for figures are provided as a Source Data file. Other data generated during and/or analyzed during the current study are available from the corresponding author on request.

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

## Acknowledgements

We thank Chandana Devi for technical support. Work in the CT lab was supported by the National Institutes of Health, National Institute of Allergy and Infectious Diseases Grants AI137541, AI125429, AI126133, AI132996 and partly by Immunome (supplementary studies Fig. S3). Work in the SD lab was supported by National Institutes of Health, National Institute of Allergy and Infectious Diseases R21 AI119368-01A1, and the Lankenau Institute for Medical Research. S.D. holds the Joseph and Ray Gordon Chair in Clinical Oncology Research, which was established by the Gordon family and the Lankenau Medical Center. Work in the NR laboratory was supported in part by the National Institutes of Health, National Institute on Aging 5U01AG010483-20.

## Author contributions

S.A.T. and R.D.P.: conceptualization, methodology, investigation, and writing. P.S.: conceptualization and investigation. L.K.N., A.L.M., and C.Q.: investigation and methodology. S.G.: supervision. N.R.: supervision and funding acquisition. B.A.B.: conceptualization, methodology, supervision, and writing-original draft preparation. S.K.D. and C.T.: conceptualization, methodology, funding acquisition, project administration, supervision, writing-original draft preparation, review, and editing.

## Competing Interests

The human mAb 3H3 has been licensed to Immunome, Inc., and SD is on the Immunome Scientific Advisory Board and holds Immunome Stock. S.D. is an inventor of the 3H3 mAb and therefore may be eligible to receive royalty payments. C.T. and S.D. are inventors on a PCT patent application based on this work. The remaining authors do not declare any competing interests.
