## [Peer Review File · Nature Communications]

Reviewers' comments:

Reviewer #1 (Remarks to the Author):

1. The introduction would benefit from the use of more recent and more pertinent references. E.g. references 1-5 are 15-20 years old. Reference 10 is a review paper and I am unsure whether it is the best for the statement that biofilms contribute to development and spread of AMR. Reference 18 and 19 are also reviews, and I would recommend to cite the appropriate primary literature concerning the role of curli in biofilm formation.
2. I suggest to make a clear distinction between resistance and tolerance (line 93).
3. How does treatment affect the number of CFU in the biofilm?
4. What is the net surface charge of the glyoxylate beads? I suspect these are negatively charged? What about accessibility to neutral or positively charged particles?
5. What is the rationale for doing in vitro experiments at 28°C?
6. What is the MIC for ampicillin, i.e. 30 µg/ml AMP much above the MIC?
7. My major comment refers to the use of a single *S. Typhimurium* strain and the use of one class of antibiotics (beta-lactams). The impact of this (already exciting!) study would be considerably higher if key results were confirmed with a small set of additional (preferably recent clinical) isolates and with additional antibiotics belonging to other classes (e.g. aminoglycosides and fluoroquinolones).
8. Line 411: Firmicutes are listed as Gram-negative bacteria – please rephrase.

Reviewer #2 (Remarks to the Author):

The authors examine the impact of amyloid antibodies on *Salmonella* biofilm formation.

Study is similar in many ways to recently published Plos Pathogens DOI: 10.1371/journal.ppat.1007978. This paper should be references.

The studies and interpretation are most clear and most of the conclusions seem justified. I do have a few suggestions to further demonstrate antibody activity/specificity. Additionally, I have a few suggestions to improve clarity in terminology.

1. Is this antibody biofilm specific i.e. does the Ab impact planktonic cells
2. Fig 1 Is there an Ab dose response?
3. Line 163: Any change in cell viability?
4. If these are mature biofilms, how are the amyloid antibodies reducing amyloid matrix mechanistically?
5. Line 187 unclear how "tightly" was assessed? Not sure this is a valid descriptor
6. Line 189, sup fig 3 - So dispersion as opposed to change in viability?
7. Line 244 Penetration was not assessed.
8. Line 245 disruption is an inexact term. Is this a change in viability, matrix quantity, dispersion?
9. So 50% reduction at a supra-physiologic ampicillin concentration? What happens at physiologic

concentrations of this antibiotic i.e. concentrations typical in patients.

10. Line 248 Salmonella are a rare cause of CLABSI

11. Throughout the manuscript text is it unclear if this is pre- or post-biofilm formation in the text. Legends do a much better job of this. Consider similar clarity in text.

12. Lines 281-83 It is not clear how supernatants from these cultures address the impact of Ab on this process, especially since the amyloid is insoluble?

13. Line 300 Can the authors speculate why amyloid fibrils exposed to this Ab would be more likely to be taken up by macrophages?

14. Lines 323-4 The binding and characterization of binding have not been demonstrated in the present studies.

15. Lines 332-3 Both the rigid, loose, and quantitative ECM descriptors were not adequately demonstrated with the current approaches. Consider removing or changing these descriptors.

MS ID#: NCOMMS-19-20471-T

MS Title: *Salmonella* Typhimurium biofilm disruption by a human antibody that binds a conserved pan-amyloid epitope on curli

We thank the editors of Nature Communications for considering our manuscript for publication. Moreover, we also thank both reviewers for their comments and suggestions. We have revised the manuscript to address the comments made by both reviewers. Within this document you will find the original reviewer comments and a point-to-point response addressing the reviewers' questions and comments in blue text. In addition, the corrections are represented in the manuscript in blue text.

Reviewer #1 (Remarks to the Author):

1. The introduction would benefit from the use of more recent and more pertinent references. E.g. references 1-5 are 15-20 years old. Reference 10 is a review paper and I am unsure whether it is the best for the statement that biofilms contribute to development and spread of AMR. Reference 18 and 19 are also reviews, and I would recommend to cite the appropriate primary literature concerning the role of curli in biofilm formation.

We apologize for this oversight. We have revised the manuscript to reflect the most recent, primary references as well as the critical review articles related this research. Please see the updated manuscript, which reflects these changes.

2. I suggest to make a clear distinction between resistance and tolerance (line 93).

This is a valid criticism. As the reviewer pointed out, the formation of biofilms contributes to the development of both tolerance and resistance. We reflected this point in the introduction.

3. How does treatment affect the number of CFU in the biofilm?

This is an important point. Our results point to a mechanism for 3H3 that effects both polymerization of curli fibrils as well as opsonization/uptake of the bacteria. We determined that treatment with 3H3 alone does not alter the number of CFU in the biofilm. Instead, treatment with 3H3 alters the architecture of the biofilm (Figure 2) creating a less compact three-dimensional structure, where more bacteria can be enumerated in the uppermost layers of the biofilm (Figure 2A-C, Figure 6A). The altered biofilm structure thus allows for greater access to macrophages and subsequent uptake of the bacteria by the macrophages (Figure 6B) and increased efficacy of antibiotics (Figure 4). The results of the treatment of biofilm with 3H3 only were not described in the previous submission because they essentially negative. But we agree that this data is relevant and we now discuss it and refer to it as "data not shown."

4. What is the net surface charge of the glyoxylate beads? I suspect these are negatively charged? What about accessibility to neutral or positively charged particles?

The beads used in this study are indeed negatively charged. Accessibility of the beads to the biofilm depends on many factors, such as the material which makes up the composition of the biofilm, interactions of the components (i.e. curli-DNA complexes) and the structural integrity of the biofilm. Although curli and DNA are both strongly anionic, they can form self-assembled structures in the presence of divalent cations like Ca²⁺ or Mg²⁺, which are both physiologically relevant in the present context. Extensive work in polyelectrolyte physics and physical chemistry has shown that the presence of divalent cations can electrostatically “glue” two like charged polymers (such as curli and DNA) together [reviewed in Wong and Pollack, 2010, Ann. Rev. Phys. Chem]. Furthermore, the presence of curli blocks the penetration of bacteriophages (which are themselves negatively charged and much smaller than beads used in this study) into the matrix and biofilm structure, preventing the lysis of bacteria. It is possible that neutral or positively charged particles may interact differently with the biofilm than negatively charged particles. Nonetheless, our data suggest that the compact and rigid structure that curli provides the biofilm matrix is the primary reason why the particles do not penetrate the untreated biofilm. Altering the topography of the biofilm with the 3H3 mAb allows beads to penetrate the matrix and move laterally within it. Currently, we have an ongoing study investigating the effects of curli on the physical properties of the biofilm matrix and this point will be fully addressed in a future manuscript.

5. What is the rationale for doing in vitro experiments at 28°C?

The environmental cues that lead to the expression of enteric biofilms *in vivo* are not fully characterized. However, in laboratory conditions, low osmolarity and low temperature are often used to induce curli expression. It must be noted that *Salmonella* can express curli *in vivo* at 37°C (Humphries et al. 2008 Infection and Immunity). Similarly *E. coli* expresses curli during urinary tract infections (Kai-Larsen et al., 2010, PLoS Path) and sepsis (Bian et al., 2000, J Infect Dis).

6. What is the MIC for ampicillin, i.e. 30 g/ml AMP much above the MIC?

30 g/ml is the reported MIC for *S. Typhimurium*. This concentration is ineffective against the *S. Typhimurium* biofilm.

7. My major comment refers to the use of a single *S. Typhimurium* strain and the use of one class of antibiotics (beta-lactams). The impact of this (already exciting!) study would be considerably higher if key results were confirmed with a small set of additional (preferably recent clinical) isolates and with additional antibiotics belonging to other classes (e.g. aminoglycosides and fluoroquinolones).

We thank the reviewer for this important suggestion. In response, we performed two additional experiments. First, we tested the ability of 3H3 (0.1 µg/mL to 500 µg/ml) to inhibit biofilm formation of various clinical isolates including *S. enteritidis*, *S. pullorum* (ATCC 9120 isolated from a patient with diarrhea) and uropathogenic *E.coli* UTI89 (isolated from a patient with an acute bladder infection) that express curli. We also tested *E.coli* MC4100, a common laboratory strain of *E.coli*. Using the crystal violet assay described in the manuscript, we observed a significant reduction in pellicle biomass when 3H3 was added to cultures of *S. enteritidis*, *E.coli* UTI89, and *E.coli* MC4100. Specifically, biofilm inhibition was observed (*p<0.05) with *S. enteritidis* when incubated with 50, 250 or 500 µg/mL 3H3 (Supplemental Figure 3A). For the *E.coli* UTI89 clinical isolate, biofilm inhibition was observed (*p<0.05) when incubated with 25, 50, 250 or 500 µg/mL 3H3 (Supplemental Figure 3B). Furthermore, inhibition was observed

(* $p < 0.05$) with *E. coli* MC4100 when incubated with 50, 250 or 500 $\mu\text{g/mL}$ 3H3 (Supplemental Figure 3C). We did not see any significant impact upon the biofilm growth of *S. pullorum* (Supplemental Figure 3D). We suspect that the conditions required to induce curli expressing biofilm formation were not achieved for this strain as the absorbance was extremely low for the untreated control. We conducted these experiments in parallel with *S. Typhimurium* and observed significant pellicle biomass reduction upon incubation with 50, 250 or 500 $\mu\text{g/mL}$ 3H3 (Supplemental Figure 3E). With *S. Typhimurium*, the most significant biofilm reduction was observed upon incubation with 500 $\mu\text{g/mL}$ 3H3. Overall, we found that 3H3 was efficacious against various clinical isolates, including *S. enteritidis* and uropathogenic *E. coli*. We have included this information as Supplemental Figure 3. See below.

Second, as suggested, we tested whether additional antibiotics (in combination with 3H3) could lead to a reduction in biofilm formation similar to what was observed with ampicillin. To do this, we tested two additional antibiotics; streptomycin (aminoglycoside) and ciprofloxacin (fluoroquinolone). Both of these have been shown to have anti-biofilm activity. We established *S. Typhimurium* biofilms in presence or absence of 3H3 for 72 hours and then added these antibiotics for an additional 24 hours, with or without the 3H3, and imaged the biofilms with confocal microscopy. We observed a significant reduction in biofilm thickness with biofilms were treated in combination with 3H3 and ciprofloxacin (0.125 $\mu\text{g/mL}$), compared to either alone (Supplemental Figure 6A&B). Although not statistically significant, there was an observable

reduction in the biomass when treated in combination with 3H3 and streptomycin (12.5 μ g/mL) (Supplemental Figure 6A&B). Importantly, neither antibiotic by itself (ciprofloxacin or streptomycin) had any activity upon biofilm thickness (Supplemental Figure 6A&B). Thus, the combinatorial effect of 3H3 and antibiotics can be seen with multiple, clinically important classes of antibiotics. We included this information as Supplemental Figure 6 within the manuscript.

8. Line 411: Firmicutes are listed as Gram-negative bacteria – please rephrase.

We thank the review for this comment and apologize for the oversight. We have corrected the manuscript and the sentence now reads “Lastly, as amyloids are widespread and found in biofilms of bacteria belonging to various phyla (Bacteroidetes, Proteobacteria, Firmicutes, and Thermodesulfobacteria)^{59 27} including important human pathogens such as *Staphylococcus aureus*⁴⁶, our findings may be generalizable to the treatment of many different bacterial biofilms.”

Reviewer #2 (Remarks to the Author):

Study is similar in many ways to recently published Plos Pathogens DOI: 10.1371/journal.ppat.1007978. This paper should be referenced.

We thank the reviewer for bringing the paper entitled Structural Insights into Curli CsgA Cross- β Fibril Architecture Inspire Repurposing of Anti-amyloid Compounds as Anti-biofilm Agents by Perov *et al.*, 2019. This paper was not cited in the original manuscript because it was not published yet when we submitted our manuscript to the Nature Communications. We are glad that these studies support our own findings and we have referenced the work in the discussion linking structural similarities CsgA of curli to human pathogenic amyloids.

The studies and interpretation are most clear and most of the conclusions seem justified. I do have a few suggestions to further demonstrate antibody activity/specificity. Additionally, I have a few suggestions to improve clarity in terminology.

We thank the reviewer for appreciating the significance of this work.

1. Is this antibody biofilm specific i.e. does the Ab impact planktonic cells

We thank the reviewer for this question. The 3H3 antibody binds specifically to amyloids, including amyloid curli. Curli is only expressed by biofilm-associated *S. Typhimurium* bacteria. Thus, as curli is not expressed by planktonic *S. Typhimurium*, the antibody has no effect on planktonic cells.

2. Fig 1 Is there an Ab dose response?

We thank the reviewer for this question. In the revised manuscript, we performed a dose response assay using multiple strains of *Salmonella* and *E. coli*. See above response to Reviewer 1 (critique 7) and Supplemental Figure 3.

3. Line 163: Any change in cell viability?

Treatment with 3H3 alone does not alter the number of CFU in the biofilm. See above response to Reviewer 1 (critique 3)

4. If these are mature biofilms, how are the amyloid antibodies reducing amyloid matrix mechanistically?

This is a very interesting and astute question. Our current hypothesis is that the antigen binding domain of the 3H3 forms a cross-beta structure that is amyloid-like, or that can assume an amyloid-like structure when it contacts the end of an amyloid strand, so that it functions as a chain terminator in elongation. In the established biofilm, our provisional hypothesis is that 3H3 binding to curli fibril ends is tight enough that steric or charge effects initiate matrix loosening. Alternatively, 3H3 binding may interfere with interactions between curli and other matrix components (the biofilm is only 85% curli), sterically forcing matrix components apart from each other. We have undertaken structural, kinetic, and functional studies of curli:3H3 binding, which will hopefully help answer this question. We left these concepts out of the manuscript because they are completely speculative at this point.

5. Line 187 unclear how "tightly" was assessed? Not sure this is a valid descriptor

We agree with the reviewer. We removed this description.

6. Line 189, sup fig 3 - So dispersion as opposed to change in viability?

Yes. That is correct.

7. Line 244 Penetration was not assessed.

We agree with the reviewer. We changed the "penetration" to "activity"

8. Line 245 disruption is an inexact term. Is this a change in viability, matrix quantity, dispersion?

We agree with the reviewer. We changed the "disruption" to "disruption of the extracellular matrix structure". There is a change in the matrix which we have demonstrated with the curli stainings in Figures 1 and 2.

9. So 50% reduction at a supra-physiologic ampicillin concentration? What happens at physiologic concentrations of this antibiotic i.e. concentrations typical in patients.

The concentration of ampicillin consumed ad libitum by the mice in vivo studies (1 mg/ml) was determined in accordance with mouse models established previously (Cummings et al., 2006, Mol Microbiol). The point of this study was to establish a mouse model to study the effects of different treatments on biofilms *in vivo*. To do this, we needed to empirically identify doses at which antibiotics are ineffective against the *S. Typhimurium* biofilm in our model system. It is well known that Ampicillin is not the first choice of drug in human patients for most Enterobacteriaceae infections, due in part to drug resistance; specific antibiotic use practices depend on the agents suspected and local epidemiology, microbial cultures and sensitivities (if available), and host factors. Therefore, our addition of ciprofloxacin and streptomycin treatments in the revised manuscript were useful to confirm the ability of 3H3 to enhance antibiotic penetration of a biofilm generally. It is also important to note that our *in vitro* experiments used a clinically relevant ampicillin concentration, 30 g/ml, which is ~6 fold lower than the mean maximum serum concentration of ampicillin achieved in pediatric patients following intravenous administration with the beta-lactamase inhibitor sulbactam.

10. Line 248 *Salmonella* are a rare cause of CLABSI

We thank the reviewer for this important critique. We changed the introduction to highlight the importance of both *Salmonella* and *E. coli* biofilms in different clinical settings. We also cited original research studies (rather than reviews) to show the significance of *E. coli* infections that are associated with biofilms, particularly prosthetic joint infections, recurrent urinary tract infections, and CLABSIs. To address if 3H3 is effective against *E. coli*, we performed assays where we grew biofilms of *E. coli* MC4100 and uropathogenic *E. coli* UTI89, a clinical isolate isolated from a patient with an acute bladder infection. We saw significant reduction in biofilm biomass when incubated with 3H3. We included this information in the Supplemental Figure 3. B and C. The introduction now reads as follows:

"Biofilms are three-dimensional multicellular communities that allow bacteria to irreversibly adhere to indwelling medical devices and provide resistance to antibiotics and eradication by innate immune cells. Biofilm-associated infections of indwelling medical devices are refractory to antibiotic treatment and require surgical debridement and/or device removal¹⁻⁸. Biofilm-producing enteric bacteria including *Salmonella enterica* and *E. coli* remain to be the major cause of many bloodstream infections^{9,10,11}. While biofilms of *Salmonella* play a critical role in persistent infections¹², *E. coli* biofilms are important causes of prosthetic joint infections, recurrent urinary tract infections, and central line associated blood infections (CLABSIs)¹³⁻¹⁷"

11. Throughout the manuscript text is it unclear if this is pre- or post-biofilm formation in the text. Legends do a much better job of this. Consider similar clarity in text.

We thank the reviewer for this comment and have went through the manuscript text thoroughly to ensure that the distinction between pre or post biofilm formation is clear.

12. Lines 281-83 It is not clear how supernatants from these cultures address the impact of Ab on this process, especially since the amyloid is insoluble?

We thank the reviewer for this question. In this experiment, we aimed to investigate if there is an impact on macrophage phagocytosis when macrophages were stimulated with biofilm supernatant that was incubated with or without 3H3. As stated on line 279-281 "We tested whether 3H3 can trigger phagocytosis of biofilm-associated bacteria by innate immune cells using bone marrow-derived macrophages in vitro." We showed that there was an increase in phagocytosis by macrophages that received biofilm supernatant that was incubated with 3H3. Phagocytosis was measured by lysing the macrophages after stimulation and counting the colony forming units from the macrophage lysate. The supernatants from biofilms treated with 3H3 address the impact of Ab on this process by showing that more bacteria are phagocytosed in the presence of 3H3. When incubated with 3H3, it is speculated that that the 3H3 enhances bacterial opsonization (see comment number 13 below). We further explored this notion by incubating curli purified from *S. Typhimurium* biofilms (labeled with congo red) in the presence or absence of 3H3 and then stimulating macrophages with the congo red labeled curli alone or congo red labeled curli incubated with 3H3 (Figure 6C). We saw that there was increased fluorescence in macrophage lysate when macrophages were stimulated with curli incubated with 3H3 than curli alone. Again, this suggests that interactions between 3H3 and curli enhance macrophage phagocytosis. This is an important finding because biofilms are normally resistant to macrophage uptake (Hernandez-Jimenez et al., 2013, *Biochem Biophys Res Commun.*)

13. Line 300 Can the authors speculate why amyloid fibrils exposed to this Ab would be more likely to be taken up by macrophages?

We speculate that amyloid fibrils exposed to the antibody 3H3 would more likely to taken up by macrophages through a process of opsonization. In the process of opsonization, the Fc portion of the antibody (3H3) bound to antigen (curli) would be expected to engage the Fc receptor on macrophages and initiate phagocytosis. We discuss this in the Discussion section, lines 389 to 390 and have revised the manuscript to again mention this concept on line 382. Furthermore, as we published previously, curli fibrils activate TLR2 receptors on macrophages. TLR2 receptors are responsible for uptake of A β amyloid specifically (Liu et al., 2012, *J Immunol.*). and they are also known to stimulate the uptake of IgG opsonized antigens (Liu et al., 2012, *J Immunol.*). We recognize that this is a speculation, however it is also beyond the scope of this manuscript.

14. Lines 323-4 The binding and characterization of binding have not been demonstrated in the present studies.

We have shown through surface plasmon resonance assays that 3H3 binds to the pan-amyloid epitope (Supplementary Figure 1 and 2) and from data previously published (Levites et al., 2015, J Neurosci.). As 3H3 inhibits polymerization of CsgA_{R4-5} (Supplemental Figure 4), the synthetic curli amyloid, it is likely that 3H3 directly interacts with curli. We have undertaken a structural and kinetic study of the 3H3:curli interaction, but this is outside the scope of the current manuscript. We agree with the reviewer that the binding kinetics were never quantitatively assessed in the manuscript, so we have changed the text to read, "Here, we describe how 3H3 interacts with the bacterial amyloid curli and inhibits polymerization of curli monomers, which correlates with profound effects on the structure and function of the biofilm *in vitro* and *in vivo*"

15. Lines 332-3 Both the rigid, loose, and quantitative ECM descriptors were not adequately demonstrated with the current approaches. Consider removing or changing these descriptors.

We agree with the reviewer. Now the statement reads:" The treated biofilm displayed changes in the extracellular matrix composition and structure, which facilitated lateral movement of the glyoxylate beads within the biofilm."

REVIEWERS' COMMENTS:

Reviewer #1 (Remarks to the Author):

-Overall I am satisfied with the response to the comments made and I think the additional experiments have made this is a stronger study.

-Material and Methods section should be updated to include experiments with additional strains and antibiotics.

-I am still a bit surprised to see that references 1-6 are 11-32 years old. This is a fast-moving field after all...

Name: Tom Coenye

Reviewer #2 (Remarks to the Author):

The authors have addressed the majority of the questions and suggestions with clarified text and in some circumstances additional data. In my view the manuscript is improved. I have a couple of remaining questions/concerns.

1. Response to reviewer 2 number 1. Was this test. Easy to do
2. Response to review 2 number 9. Yes, but beta-lactams exhibit time dependent killing thus short period at a high concentration is not relevant. The authors should note this as a limitation or demonstrate activity at much lower concentrations.

MS ID#: NCOMMS-19-20471-T

MS Title: Salmonella Typhimurium biofilm disruption by a human antibody that binds a pan-amyloid epitope on curli

We thank the editors of Nature Communications for considering our manuscript for publication. Moreover, we also thank both reviewers for their comments and suggestions. We have revised the manuscript to address the comments made by both reviewers.

REVIEWERS' COMMENTS:

Reviewer #1 (Remarks to the Author):

-Overall I am satisfied with the response to the comments made and I think the additional experiments have made this a stronger study.

We thank the reviewer for appreciating the significance of this work.

-Material and Methods section should be updated to include experiments with additional strains and antibiotics.

The new experiments and the information on bacterial strains was included in the supplementary information. However, this information is now moved into the main Methods section of the manuscript.

-I am still a bit surprised to see that references 1-6 are 11-32 years old. This is a fast-moving field after all...

References 1-6 are associated with the original description of biofilms and some critical characteristics that are relevant to this study. We wanted to give credit to the original studies that have shaped this field and therefore included these seminal studies. However, upon reviewer's recommendation, we have included some of the newer references as well. Please see the updated manuscript, which reflects these changes to the references.

Reviewer #2 (Remarks to the Author):

The authors have addressed the majority of the questions and suggestions with clarified text and in some circumstances additional data. In my view the manuscript is improved. I have a couple of remaining questions/concerns.

1. Response to reviewer 2 number 1. Was this test. Easy to do

We tested the antibody on the planktonic cells of S. Typhimurium. We did not see any effects of the antibody on bacteria. We attributed this for curli not being present in planktonic S.

Typhimurium. Our readout was cell viability. However, it is possible that there may be some effects of the antibody on the bacteria that we were not able to observe. More detailed analysis may be required to test this.

2. Response to review 2 number 9. Yes, but beta-lactams exhibit time dependent killing thus short period at a high concentration is not relevant. The authors should note this as a limitation or demonstrate activity at much lower concentrations.

This is a valid criticism. We included this point as a limitation in the discussion.